# SemiSANet: A Semi-Supervised High-Resolution Remote Sensing Image Change Detection Model Using Siamese Networks with Graph Attention

Chengzhe Sun [1], Jiangjiang Wu [1], Hao Chen [1,2,*] and Chun Du [1]

[1] Department of Cognitive Communication, College of Electronic Science and Technology, National University of Defense Technology, Changsha 410073, China; sunchengzhe@nudt.edu.cn (C.S.); wujiangjiang@nudt.edu.cn (J.W.); duchun@nudt.edu.cn (C.D.)

[2] Key Laboratory of Natural Resources Monitoring and Supervision in Southern Hilly Region, Ministry of Natural Resources, Changsha 410083, China

[*] Correspondence: hchen@nudt.edu.cn

**Abstract:** Change detection (CD) is one of the important applications of remote sensing and plays an important role in disaster assessment, land use detection, and urban sprawl tracking. High-accuracy fully supervised methods are the main methods for CD tasks at present. However, these methods require a large amount of labeled data consisting of bi-temporal images and their change maps. Moreover, creating change maps takes a lot of labor and time. To address this limitation, a simple semi-supervised change detection method based on consistency regularization and strong augmentation is proposed in this paper. First, we construct a Siamese nested UNet with graph attention mechanism (SANet) and pre-train it with a small amount of labeled data. Then, we feed the unlabeled data into the pre-trained SANet and confidence threshold filter to obtain pseudo-labels with high confidence. At the same time, we produce distorted images by performing strong augmentation on unlabeled data. The model is trained to make the CD results of the distorted images consistent with the corresponding pseudo-label. Extensive experiments are conducted on two high-resolution remote sensing datasets. The results demonstrate that our method can effectively improve the performance of change detection under insufficient labels. Our methods can increase the IoU by more than 25% compared to the state-of-the-art methods.

**Keywords:** change detection; remote sensing; semi-supervised learning; data augmentation; consistency regularization

## 1. Introduction

The change detection (CD) task performed on remote sensing (RS) images aims to identify semantic changes that occur in remote sensing images of the same areas acquired at different times [1]. Due to the increasing number of Earth observation programs such as Sentinel, WorldView, GeoEye, ZY-3, and GF, various sensors provide a large number of remote sensing images with different resolutions for change detection tasks. At the same time, with the development of deep learning (DL) technology, the change detection methods nowadays can quickly obtain the change information of the area we are concerned with. So far, remote sensing image change detection technology has been widely used in the fields of ecosystem monitoring [2], land cover and land use mapping [3], damage assessment [4], and urban expansion monitoring [5].

Research on remote sensing image change detection techniques has been conducted in the remote sensing community for decades. Traditional change detection methods can be divided into three classes [6]: (1) image arithmetical based, (2) image transformer based, and (3) post classification based. Image arithmetical-based methods first produce difference images by comparing pixel-values from multi-temporal images, and then use

thresholds to classify pixels into a changed class or unchanged class. Image transformation-based methods such as principal component analysis (PCA) discriminate the changed pixels by transforming image spectral combinations into a specific feature space. Post classification-based methods first classify the bi-temporal images independently. Then the change detection results can be obtained by directly comparing the classification results [7].

Currently, the main change detection algorithms are fully supervised methods based on deep learning, which requires a large number of data to reach high accuracy. In remote sensing image CD tasks, data are the bi-temporal RS images, and their labels are the change pixels labeled artificially. When making a label of a pair of bi-temporal images, we should first identify the changed areas with our eyes, and then fill the changed and unchanged areas with different pixel values, such as 255 and 0. It is obvious that making labels for thousands of bi-temporal images will cost huge labor and time. This limitation makes the fully supervised methods difficult to be widely applied in the field of remote sensing image change detection.

Normally, there are three ways to overcome this limitation. The first one is to increase the amount of labeled data by data augmentation such as rotating, shifting, cropping, and flipping. This method enables improvement of the accuracy in the fully supervised CD task, but may not perform well on unseen datasets since the enlarged images are the variation of the original images. The second one is weakly supervised learning [8], where the data is coarsely annotated by image-level labels [9], bounding boxes [10], and graffiti [11]. In general, the coarsely labeled information can help generate pseudo-labels to further guide the supervised learning of neural networks. Weakly supervised methods save some labor and time compared to fully supervised methods. However, for complex areas CD tasks in high-resolution remote sensing images, it still costs a lot of effort to identify and coarsely label changes artificially. The third method is semi-supervised learning (SSL) [12]. Different from the previous two methods, semi-supervised learning aims to use a small amount of labeled data and a large amount of unlabeled data to train models. It can significantly reduce the labor and time spent on labeling, and exploit the feature of unlabeled data to avoid over-fitting at the same time. In the field of remote sensing, the number of images is huge but it is hard to make labels. It is natural to use unlabeled data to improve the performance of remote sensing image processing. Semi-supervised learning can be achieved by generative adversarial networks (GAN) [13], autoencoder [14], and graph neural networks [15]. In the field of remote sensing, semi-supervised learning has been used in wildfire fuel mapping [16,17], hyperspectral image classification [18], and semantic segmentation [19].

However, only a few studies have been conducted to deal with semi-supervised remote sensing image change detection [20]. For example, Peng et al. [20] proposed a semi-supervised CD method based on generative adversarial networks (GAN). By constructing two discriminators to strengthen the feature distribution consistency of detection results between labeled data and unlabeled data, the model performance is improved by utilizing a large amount of unlabeled data. This work successfully achieves semi-supervised change detection, but since the training of the GAN is not stable and the hyperparameters are difficult to adjust, this method is a bit complicated to apply in practice. Additionally, the performance of change detection in case of insufficient labeled data can be improved.

To address this limitation, a simple and effective semi-supervised change detection method is investigated in this paper. By simply training the model to make consistent change detection results for unlabeled images with and without the strong augmentation, our method can improve the anti-interference robustness of the change detection model. This leads to a high change detection accuracy in case of insufficient labeled images. At the same time, to achieve better performance of change detection, we also improve the model by using a graph attention mechanism (GAT). The main contributions of this paper are as follows:

(1)    A novel semi-supervised RS image change detection framework based on pseudo-label and consistency regularization is proposed. Firstly, labeled data is employed

to train a network and with the help of a pixel-level threshold filter, high-confidence pseudo-labels for unlabeled RS images are generated. Then, by making the change detection results of distorted RS images consistent with the pseudo-label, the performance and robustness of our model are improved.

(2) A novel RS image change detection network based on the Siamese Unet is proposed, and the addition of the graph attention mechanism gives the model the ability to extract long-range dependencies between latent features.

(3) Extensive experimental results on two high-resolution remote sensing datasets demonstrate that our model can greatly improve change detection results in the case of an insufficient number of labels and outperform the state-of-the-art change detection methods. The optimal range of hyperparameter values for this semi-supervised CD method is given by extensive ablation experiments.

## 2. Related Work

In this section, the DL framework for CD, the attention mechanism, and the semi-supervised method based on consistency regularization will be briefly illustrated.

### 2.1. DL Framework for CD

In recent years, with the advent of the era of big data and the rapid development of computing power, deep learning (DL) algorithms have become a research hotspot [21]. Deep learning can learn multiple levels of representation and abstraction to help understand images [22], sounds [23], and texts [24] and extract semantic information from them. It has much more powerful modeling capabilities than traditional algorithms and has achieved many successes in artificial intelligence fields such as image processing, language recognition, and natural language processing. In the field of remote sensing images processing, deep learning has also shown excellent performance [25] and is widely used in problems such as image registration [26], mutual superimposing [27], road extraction [28], image-to-map translation [29,30], and image segmentation [31]. Deep learning is also an effective solution to the remote sensing image change detection problem [32]. Benefiting from remote sensing big data, many CD algorithms based on fully supervised learning have been proposed. Most of the DL networks for CD tasks are based on convolutional neural networks (CNN). Among them, the UNet based on fully convolutional networks is the most popular and has become one of the standard CNN architectures for CD tasks with many extensions [32]. UNet is a symmetric encoder–decoder structure that captures contextual information to extract features in the downsampling part and reconstructs the image and outputs the final change detection results in the upsampling part [33]. By adding a skip connection between encoder and decoder, UNet can better integrate deep semantic information and shallow spatial information, improving the accuracy of CD. Due to the characteristic that the inputs of CD tasks are bi-temporal RS images, Daudt et al. [34] combined the Siamese structure into UNet, making the encoder part a dual-channel structure with shared parameters. In this work, the bi-temporal remote sensing images are first processed by downsampling respectively, then the extracted features will be integrated together by difference or connection as the input of the decoder. This improvement better guides the network to compare differences between images. Shao et al. [35] achieved change detection of satellite images and UAV images of different sizes by modifying the encoder part of UNet to adjust images of different sizes. Peng et al. [36] introduced the Nested UNet (UNet++) structure into the CD task by adding more intermediate (nested) convolutional blocks and skipping connections in UNet, making the connected features have higher semantic similarity, which eliminates the semantic gap between encoder and decoder. Fang et al. [37] combined the previous two works and proposed a Siamese network based on Nested UNet, which alleviated the loss of deep localization information in the neural network.

## 2.2. Attention Mechanism

The attention mechanism originated from the human visual system [38] and has been a research hotspot in the field of image processing. It enables the network to fuse different features more effectively, and has a significant effect on improving the accuracy of image processing problems. Zhang et al. [6] introduced the channel attention mechanism and spatial attention mechanism in the decoder part of the network, which makes the network use information more effectively during the upsampling process. The channel attention mechanism learns the contribution of different channels to the CD results through a multi-layer perceptron (MLP) network, and determines the attention assigned to each channel according to the contribution. The spatial attention mechanism learns the contribution of different positions through a $7 \times 7$ convolution block. To fully exploit the spatiotemporal correlations between individual pixels at different locations and times, Chen et al. [39] proposed a spatiotemporal attention mechanism. The spatiotemporal attention mechanism is inspired by the self-attention mechanism [40], which utilizes three different convolutional layers to compute Query, Key, and Value tensors from the input feature tensors. The output vector is obtained by mapping these three vectors through calculation. The process of obtaining the output vector makes full use of the spatial and temporal dependencies between pixels, and can obtains illumination-invariant and misregistration-robust features.

The graph attention (GAT) mechanism [41] is a graph-based attention method with powerful long-range feature aggregation capabilities. In recent years, there have been some related achievements in the application of graph network methods in the field of remote sensing [31,42]. In [31], Zi et al. proposed an image segmentation method based on GAT and self-attention, which treats the features extracted by the encoder part as nodes of the graph and then uses GAT to fuse these features. The long-distance information aggregation ability of the graph can make the relationship between features be mined and utilized.

## 2.3. Semi-Supervised Method Based on Consistency Regularization

Semi-supervised learning (SSL) is a branch of deep learning which is conceptually between fully supervised learning and unsupervised learning. It aims to use a large amount of unlabeled data combined with a small amount of labeled data to accomplish specific learning tasks [12]. Semi-supervised learning methods based on consistency regularization are a class of SSL and are popular these years. The main assumption of the consistency regularization principle is that the output of the model should be consistent before and after the input or the model is perturbed. Consistency regularization can be measured by the following equation:

$$D[p_{model}(y|Augment_1(x),\theta), p_{model}(y|Augment_2(x),\theta)] \tag{1}$$

where $D$ is the metric function, which can be cross-entropy loss, mean square loss, etc. *Augment* is the data augmentation function, which will perturb the input. $p_{model}$ is the model and $\theta$ is the model parameter.

In the semi-supervised approach, the augmentation method distorts the image. The consistency regularization principle forces the model to make the same prediction for the same image with different distortions, thus enhancing the robustness and generalization of the model. Based on the principle of consistency regularization, Laine et al. [43] proposed the Π-Model. In each epoch of the training process, the same unlabeled data will be propagated forward twice, and it will go through stochastic augmentation before entering the network, resulting in two different predictions. The Π-Model improves the robustness of the model by training to narrow the differences between these pairs of predictions. An improved model, temporal ensembling, is also proposed in [43]. In temporal ensembling, the unlabeled data are propagated forward only once in each epoch, and the predictions obtained from the previous epochs are used as the output pseudo-label after exponential moving average (EMA), which improves the speed of the model. Tarvainen et al. [44] proposed the Mean Teacher model where the idea of the exponential moving average is

applied to the model itself. Specifically, there is a student model whose parameters are updated by the backpropagation algorithm, and a teacher model whose parameters are updated by the exponential moving average of student model parameters in previous epochs. Wang et al. [19] successfully achieve semi-supervised remote sensing image semantic segmentation by using Mean Teacher. To further improve the performance of semi-supervised learning networks, Berthelot et al. [45] proposed MixMatch by combining the principle of minimizing entropy. The minimizing entropy principle means that the model should make low-entropy predictions on unlabeled data. Pseudo-label is an application of minimizing entropy principle. MixMatch makes pseudo-labels by averaging and sharpening the predictions of the input unlabeled data, which is performed by multiple data augmentation. MixMatch achieves better results than previous semi-supervised methods. To make the perturbation more drastic, Xie et al. [46] introduced strong augmentation, which can cause dramatic changes in the shape and color of the image, to process the input data and proposed a UDA model. Pseudo-labels were obtained by sharpening the predictions of input data after weak augmentation such as flipping and shifting. Then, the same data was performed strong augmentation and fed into the network for training. A performance beyond MixMatch is achieved by the UDA model. Sohn et al. [47] proposed FixMatch to strengthen pseudo-labels on the basis of the UDA model. Instead of sharpening the predictions, FixMatch makes pseudo-labels by selecting predictions beyond the threshold, which effectively prevents false pseudo-labels from misleading the model.

The above semi-supervised methods based on consistency regularization have been able to achieve satisfactory accuracy in the image classification domain. However, the images used for image classification have a small number of objects, e.g., a picture may contain only a cat, a few apples, and so on. In contrast, remote sensing images have much richer semantic information. This means that it is easier to destroy the available semantic information when perturbing the remotely sensed images, which may lead to poor CD results. How to effectively use consistency regularization to implement semi-supervised learning-based CD tasks is a problem worth investigating.

## 3. Methods

In this section, the architecture of the proposed semiSANet will be illustrated first. Then, we will present details on the SANet. Finally, the loss functions of the supervised part and unsupervised part in our semiSANet will be defined.

The architecture of our proposed semi-supervised CD method is illustrated in Figure 1. It consists of two parts, the supervised part and the unsupervised part. In the supervised part, bi-temporal remote sensing images with labels are fed into the SANet after weak augmentation, and the predictions $\hat{Y}_l$ produced are subjected to a loss function calculation with binary labels $Y_l$ to obtain the supervised losses $L_s$. In the unsupervised part, the unlabeled data are fed into the model to get the pseudo-label $\hat{Y}_{pl}$ and the prediction $\hat{Y}_u$. First, weak augmentation is performed on the unlabeled data, including random flipping and shifting to obtain $X_w$. After that, $X_w$ will follow two processing streams. In the first processing stream (green arrow in Figure 1), $X_w$ is fed into the model to obtain the binary image $\hat{Y}_w$. $\hat{Y}_w$ is then subjected to strong augmentation and a confidence threshold filter to obtain the pseudo-label $\hat{Y}_{pl}$. In the other processing stream (yellow arrow in Figure 1), $X_w$ is first subjected to strong augmentation to get $X_s$ and then fed into the model to get the prediction $\hat{Y}_u$. Note that the two strong augmentations used here are identical in order to ensure the pixel-by-pixel correspondence of the pseudo-label $\hat{Y}_{pl}$ and the prediction $\hat{Y}_u$. The unsupervised loss $L_u$ is calculated using the pseudo-label and predicted $\hat{Y}_u$. The final loss $L_{CD}$ is obtained by weighted summation of supervised loss $L_s$ and unsupervised loss $L_u$. The network parameters can be updated by back-propagating using $L_{CD}$.

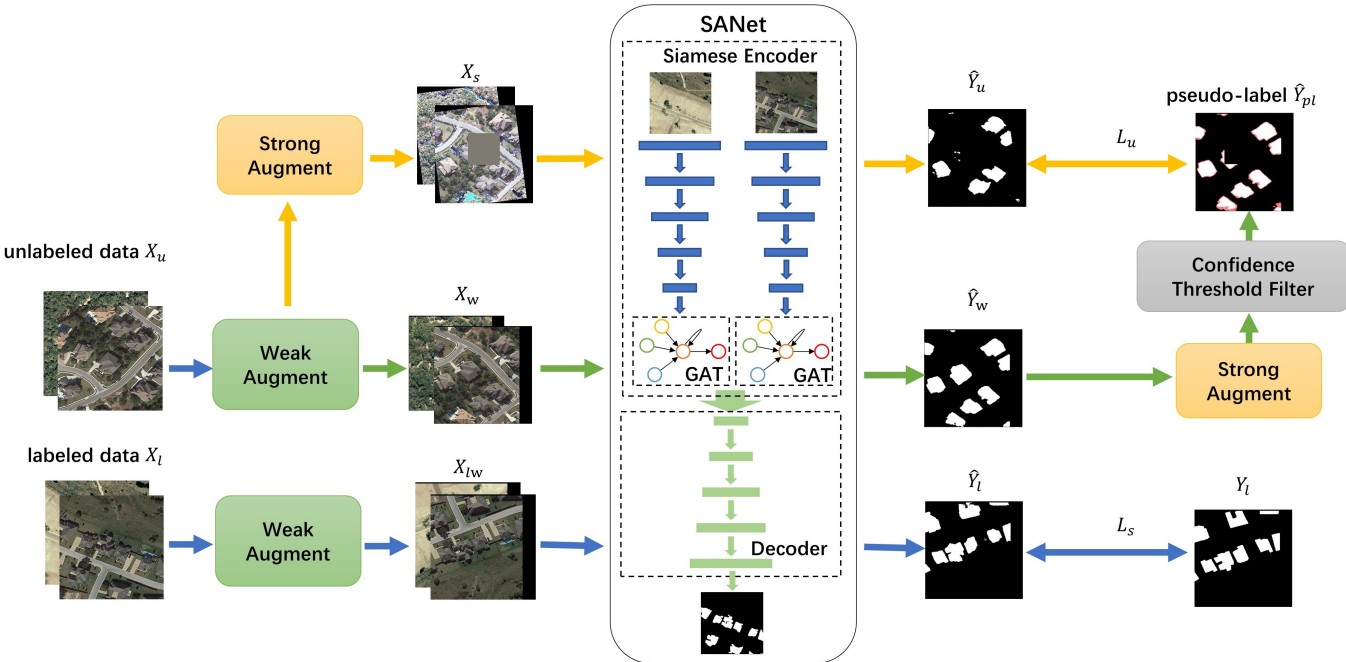

**Figure 1.** Flowchart of the method semiSANet. The labeled data are directly passed through the SANet after weak augmentation to obtain the fully supervised loss $L_s$. The unlabeled data are fed into the SANet after weak augmentation (to obtain $\hat{Y}_w$) and the confidence threshold filter to obtain the pseudo-label $\hat{Y}_{pl}$, at the same time using the strong augmentation to produce distorted image and obtain CD results $\hat{Y}_u$ by the SANet. The unsupervised loss $L_u$ is calculated by $\hat{Y}_u$ and $\hat{Y}_{pl}$.

## 3.1. Semi-Supervised Method Architecture

Our semi-supervised process can be shown more succinctly by the following equation:

$$D_l(Y_l, P_{SANet}(\hat{Y}|Aug_w(X_l), \theta)) \tag{2}$$

$$D_u(CTF(Aug_s(P_{SANet}(\hat{Y}_w|Aug_w(X_u), \theta))), P_{SANet}(\hat{Y}_u|Aug_s(X_u), \theta)) \tag{3}$$

The first equation represents the supervised part and the second one represents the unsupervised part. $\theta$ and $P_{SANet}$ represent the parameters and predictions of SANet respectively. $Aug_w$ and $Aug_s$ represent the weak augmentation and strong augmentation respectively. CTF represents the confidence threshold filter, and $D_l$ and $D_u$ represent the loss functions used for supervised and unsupervised part respectively.

### 3.1.1. Data Augmentation in semiSANet

The data augmentations used in semiSANet can be divided into weak augmentation and strong augmentation. Weak augmentations include random flipping and shifting which are used for expanding the datasets.

The strong augmentation used in this work was improved from the work Randaugment by Cubuk [48]. While using the Randaugment strategy for data augmentation, we added and removed according to the characteristics of remote sensing images, and finally obtained a total of 15 strong augmentation methods. Table 1 lists the strong enhancement methods this work used, their intensity ranges, and their specific explanations.

**Table 1.** Strong augmentation method, degree and description.

| Augmentation | Degree | Description |
| --- | --- | --- |
| Brightness | [0.05, 0.95] | Control the brightness of an image |
| Color | [0.05, 0.95] | Adjust the color balance of an image |
| Contrast | [0.05, 0.95] | Control the contrast of an image |
| Equalize | / | Equalize the image histogram |
| Identity | / | Return the original image |
| Posterize | [4, 8] | Reduce the number of bits for each color channel |
| Rotate | [−30, 30] | Rotate the image |
| Sharpness | [0.05, 0.95] | Adjust the sharpness of an image |
| ShearX | [−0.3, 0.3] | Shears the image along the horizontal axis |
| ShearY | [−0.3, 0.3] | Shears the image along the vertical axis |
| Solarize | [0, 256] | Invert all pixel values above a threshold |
| TranslateX | [−0.3, 0.3] | Translates the image horizontally |
| TranslateY | [−0.3, 0.3] | Translates the image vertically |
| Cutout | [0.25, 0.35] | Cutout an area of an image |

Figure 2 shows the results of various strong augmentation methods after acting on the remote sensing images. When using strong augmentation for remote sensing images, we randomly select four augmentations from the methods (cutout is always selected) listed in Table 1 and form a strong augmentation method, and the effect is shown in Figure 3.

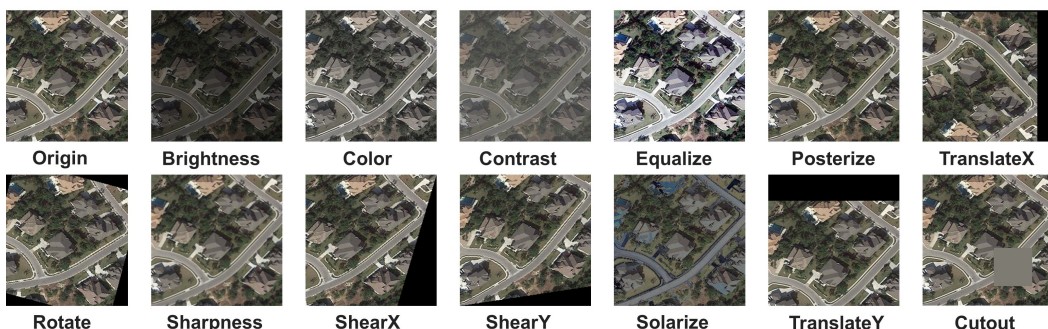

**Figure 2.** Strong augmentations used in remote sensing images.

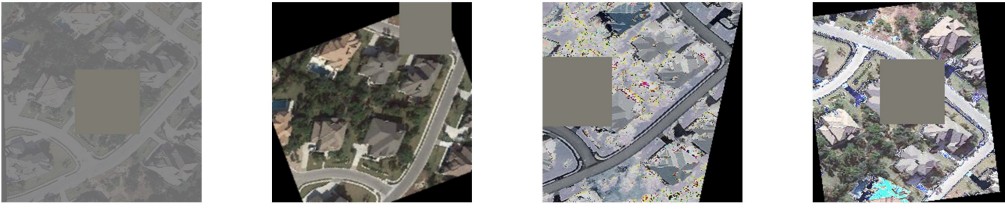

**Figure 3.** Multiple strong augmentation methods are randomly applied to remote sensing images.

It should be noted that the degree of each augmentation is random, and the degree range is listed in Table 1. When training the network, we use different strong augmentations for different pairs of bi-temporal remote sensing images at the same epoch, and for the same pair of images at different epochs to ensure that the strong augmentations are sufficiently

random. However, it should be noted that the same strong augmentation should be used for the same pair of bi-temporal remote sensing images and their pseudo-label at the same epoch to ensure that the pixels of pseudo-label $\hat{Y}_{pl}$ and unsupervised prediction $\hat{Y}_u$ can correspond to each other.

The strong augmentations serve two purposes. One is to increase or decrease the color gap between remote sensing images by color augmentation. The second is to deform the features through shape augmentation. Consistency regularization requires the model to ignore these changes and focus on object changes, improving the semantic understanding ability and robustness of the model.

### 3.1.2. Confidence Threshold Filter

The confidence threshold filter is referenced from FixMatch [47] and aims to use threshold $\tau$ to filter out the pseudo-labels with low confidence. For the characteristics of the change detection task, we filter the pseudo-labels at the pixel level. Pixels smaller than threshold $\tau$ are masked, as shown in Figure 4. When computing the unsupervised loss $L_u$ using the pseudo-label $\hat{Y}_{pl}$ and the prediction $\hat{Y}_u$, the masked pixels are not involved in the calculation. Only pixels above the confidence threshold $\tau$ are involved in the calculation of the unsupervised loss $L_u$, which helps prevent the model from being poorly trained due to incorrect pixel-level pseudo-labels.

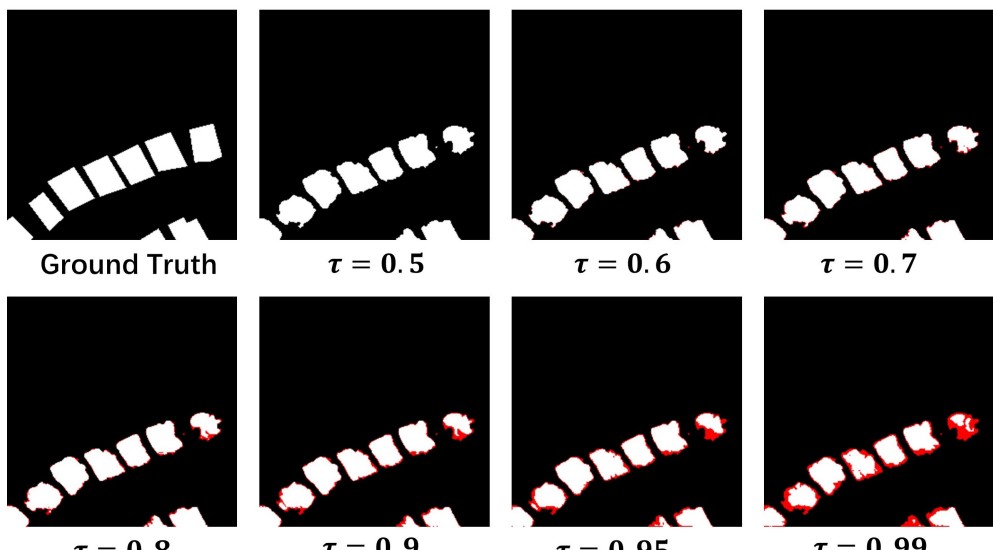

**Figure 4.** The red part is masked because it is smaller than $\tau$ and does not participate in the loss calculation.

### 3.2. SANet

SANet is an end-to-end encoder–decoder network used for CD in semiSANet. It has a UNet backbone with a dense skip connection between the encoder and decoder, which is able to maintain high-resolution features and fine-grained localization information to efficiently extract multi-scale information. Unlike UNet, we use the Siamese network as the encoder part of SANet in order to adapt to the characteristics of the CD task. The bi-temporal images are fed into the two FCN branches of the Siamese network respectively, as shown in Figure 5.

Restricted by the size of the convolutional kernel, CNN cannot fuse long-range dependency of features, thus we use the graph attention mechanism in our model. We construct the graph using the features extracted from the full convolutional network. We use features as nodes which denoted by $\{\vec{h}_1, \vec{h}_2, \vec{h}_3, \cdots, \vec{h}_N\}$ and $N$ denotes the number of features. The weight of the edge between node $i$ and node $j$, as well as the attention coefficient between $\vec{h}_i$ and $\vec{h}_j$, is denoted by $a_{ij}$. Through the graph attention mechanism, we can fuse

the information of neighboring nodes to $\vec{h}_i$, as shown in Figure 5b. First, we calculate the attention coefficient $a_{ij}$ as follows:

$$e_{ij} = LeakyReLU(\vec{a}^T[\mathbf{W}\vec{h}_i||\mathbf{W}\vec{h}_j]) \tag{4}$$

$$a_{ij} = Softmax_j(e_{ij}) = \frac{exp(e_{ij})}{\sum_{k \in N_i} exp(e_{ik})} \tag{5}$$

where $\vec{a}$ and $\mathbf{W}$ are trainable parameters, and we can achieve automatic attention allocation by neural network training. $||$ denotes the concatenate operation.

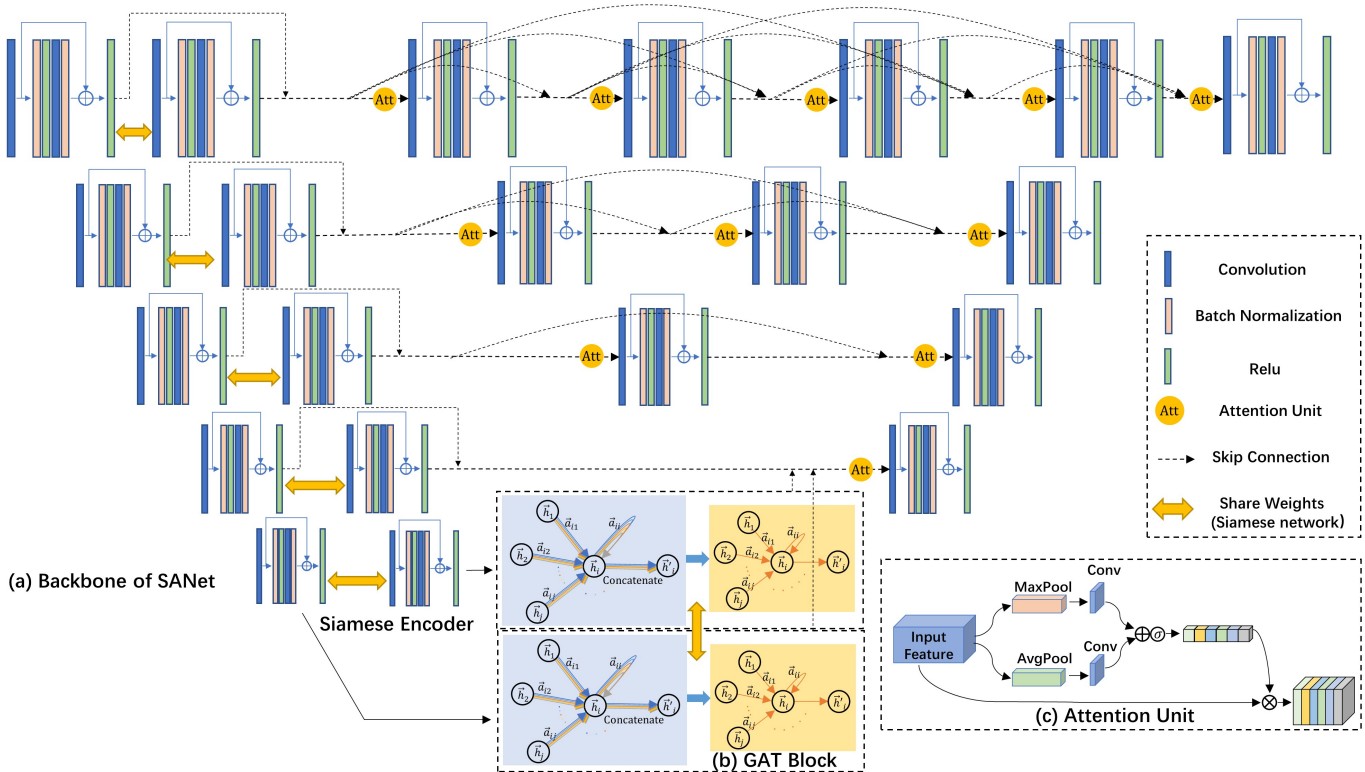

**Figure 5.** Illustration of SANet. (**a**) The backbone of SANet. (**b**) GAT block, including a multi-headed graph attention unit and a single-headed graph attention unit. (**c**) Attention unit.

Then by weighted summation we can get the new feature $\vec{h}'_i$ of node $i$. In this process, node $i$ fuses the features of its neighboring nodes. Let $N_i$ denotes the union of the indexes of the neighboring nodes of $i$, which also includes $i$, $\vec{h}'_i$ is calculated as follows:

$$\vec{h}'_i = \sum_{j \in N_i} a_{ij}\mathbf{W}\vec{h}_j \tag{6}$$

As shown in Figure 5b, the first GAT layer we use in the model is a GAT with multi-head attention, which can improve the generalization ability of the attention mechanism. Using $K$ sets of attention layers that are independent of each other, their results are concatenated together. The formula for computing $\vec{h}'_i$ using the multi-headed attention mechanism is as follows:

$$\vec{h}'_i = ||_{k=1}^{K} \sum_{j \in N_i} a_{ij}^k \mathbf{W}^k \vec{h}_j \tag{7}$$

Since the number of features directly determines the size of the graph and further determines the number of parameters, limited by GPU memory, we only add the graph attention mechanism to the end of the encoder and construct the graph using the deepest

semantic features extracted. In order to make the feature information and the spatial context information extracted by the encoder correspond to the same position of the bi-temporal remote sensing images, both channels of the Siamese network, including the GAT network, share the weights. Further, in order to suppress the semantic gap between the high-level and the low-level features, we adopt the dense attention connection mentioned in [20], which adds the channel attention mechanism to the dense skip connection between encoder and decoder. The channel attention mechanism can assign weights to different channels when connecting deep semantic information and shallow spatial information, and suppress the semantic gap between deep and shallow information.

### 3.3. Loss Function

For the CD task where the number of unchanged pixels is much larger than the number of changed pixels, we use a combination of cross-entropy loss $L_{ce}$ and dice loss $L_{dice}$ as the loss function.

Let the label be Y and the CD result $\hat{Y} = \hat{y}_k, k = 1, 2, \ldots, H \times W$, where $H$ is the height of the image, $W$ is the width of the image, and $\hat{y}_k$ is the pixel value of the $k$th pixel of the image. Let $c$ be 0 or 1, representing the change or not of the $k$th pixel in the label. The cross-entropy loss and the dice loss are calculated as follows:

$$L_{ce} = \frac{1}{H \times W} \sum_{k=1}^{H \times W} log\left( \frac{exp(\hat{y}_{kc})}{\sum_{l=0}^{1} exp(\hat{y}_{kl})} \right) \tag{8}$$

$$L_{dice} = 1 - \frac{2 \cdot Y \cdot softmax(\hat{Y})}{Y + softmax(\hat{Y})} \tag{9}$$

Let $\hat{Y}_{pl}$ denote the pseudo-label, the supervised loss $L_s$ and unsupervised loss $L_u$ are defined as follows:

$$L_s = L_{ce}(\hat{Y}_l, Y_l) + L_{dice}(\hat{Y}_l, Y_l) \tag{10}$$

$$L_u = L_{ce}(\hat{Y}_u, \hat{Y}_{pl}) + L_{dice}(\hat{Y}_u, \hat{Y}_{pl}) \tag{11}$$

The final loss of the whole semi-supervised method is obtained by weighted summation of $L_s$ and $L_u$. Let the weighting factor be $\lambda$, and the final loss is calculated as follows:

$$L_{CD} = L_s + \lambda L_u \tag{12}$$

Obviously, $\lambda$ determines the influence degree of unsupervised loss on the model, and when $\lambda = 0$, the model degenerates to a fully supervised model.

## 4. Experiments

### 4.1. Datasets

To verify the effectiveness of this method, we selected two remote sensing image CD datasets: the LEVIR dataset and the WHU dataset, both of which are VHR datasets and contain a large number of images.

- The LEVIR dataset (https://justchenhao.github.io/LEVIR/, accessed on 3 May 2022) is a large-scale remote sensing building change detection dataset consisting of 637 Google Earth VHR images of size $1024 \times 1024$, each with a resolution of 0.5 m/pixel. The dual-temporal images of the LEVIR dataset come from 20 different areas in several cities in Texas, USA, and were taken from 2002 to 2018. The dataset focuses on land use changes, primarily in buildings, and covers various types of buildings such as cottage homes, high-rise apartments, small garages, and large warehouses. The images have many pseudo-changes caused by seasons and lighting, and the building-induced changes vary in shape and size, making it a very challenging dataset. Due to the limitation of GPU memory, we segment the image pairs into $256 \times 256$ pixels and

then divide them randomly into a training set and a test set. The training set contains 7120 images while the test set contains 1024 images.

- The WHU dataset (https://study.rsgis.whu.edu.cn/pages/download/building_dataset.html, accessed on 3 May 2022) contains two remote sensing images and their change map taken in the same area of Christchurch, New Zealand in 2012 and 2016. Each image is 32,507 × 15,345 pixels in size with a resolution of 0.075 m/pixel, and the main change of interest is buildings. Due to GPU memory limitation, as with the LEVIR dataset, we segmented it by 256 × 256 pixels to obtain a training set with 2000 image pairs and a test set with 996 image pairs. The WHU dataset is a difficult dataset because of the large variation and extremely heterogeneous distribution of the change objects.

### 4.2. Evaluation Metrics

To effectively evaluate the CD effectiveness of various methods, we used F1-score, overall accuracy (OA), Kappa coefficient (Kappa), and intersection-over-union (IoU) as evaluation metrics, and they are defined as follows:

$$F1 = \frac{2 \times P \times R}{P + R} \tag{13}$$

$$OA = \frac{TP + TN}{TP + TN + FP + FN} \tag{14}$$

$$Kappa = \frac{OA - PRE}{1 - PRE} \tag{15}$$

$$IoU = \frac{TP}{TP + FP + FN} \tag{16}$$

Let TP denote the number of true positives, FP denote the number of false positives, TN denote the number of true negatives, and FN denote the number of false negatives, we can calculate P, R, and PRE as follows:

$$P = \frac{TP}{TP + FP} \tag{17}$$

$$R = \frac{TP}{TP + FN} \tag{18}$$

$$PRE = \frac{(TP + FN) \times (TP + FP) + (TN + FP) \times (TN + FN)}{(TP + TN + FP + FN)^2} \tag{19}$$

### 4.3. Experimental Setting

The proposed method is implemented in the Pytorch framework, which is installed with Intel Core i7-10700 CPU (2.9 GHz, 8 cores, and 16 GB RAM) and a single NVIDIA GTX 3090 GPU with 24GB RAM. The AdamW optimizer is adopted with the learning rate $1 \times 10^{-3}$ and the weight decay $1 \times 10^{-2}$. The proposed semi-supervised method was trained for 2000 iterations both for the LEVIR dataset and WHU dataset. Let the proportion of labeled data be $k$, the batch size is set to 4 for the labeled data and set to $\frac{4}{k} - 1$ for the unlabeled data. Additionally, the hyperparameter $\lambda$ was set as 8 and the confidence threshold $\tau$ is set to 0.95. It is important to note that we will discuss the optimal range of $\lambda$ and $\tau$ in Sections 4.5.2 and 4.5.3.

For SANet, the number of convolution kernels of each convolution module is set to {8, 16, 32, 64, 128} and the size of convolution kernels is set to 3 × 3.

Since the training set of the LEVIR dataset has 7120 pairs of dual-temporal remote sensing images, which is much larger than the actual need to obtain high precision training results. Therefore, we only randomly select 2000 pairs from it as the training set for experimental use, which is called the LEVIR2000 dataset. It is worth noting that the

other image pairs in the training set will be used in the experiments to test the effect of unsupervised data volume on the accuracy of the semi-supervised CD task.

To prove the effectiveness of our method, we choose 2.5% as the proportion of labeled images. However in application, we suggest choosing a higher proportion of labeled images to get better performance. Therefore, we also set the other three higher proportions in our experiments to show the influence of more labeled images. Finally, the proportion of labeled data was set to {2.5%, 5%, 10%, 20%} for both the LEVIR2000 dataset and the WHU dataset.

### 4.4. Baselines and Comparison

To verify the effectiveness of our method, we compare the proposed method with the state-of-the-art fully supervised CD and semi-supervised CD methods, which are as follows:

- **FC-Conc** [34] **and FC-Diff** [34]: FC-Conc and FC-Diff are the baseline methods for CD tasks, among which FC-Conc and FC-Diff are more effective, so they are cited as comparison methods in this paper. They are both based on the combination of UNet and Siamese networks. The difference is that in the deep feature fusion part, FC-Conc uses concatenation while FC-Diff uses difference.
- **SNUNet-ECAM** [37]: SNUNet-ECAM is a combination of Nested UNet and Siamese network with strong feature extraction and semantic information fusion capabilities. The ensemble channel attention mechanism (ECAM) is also incorporated, which enables the network to effectively fuse multi-level output information at the output side and achieve high accuracy in the CD task.
- **s4GAN** [49]: s4GAN is used for semi-supervised semantic segmentation. The semi-supervised approach is implemented using unsupervised data by setting the feature matching loss to minimize the difference between the predicted and true segmentation maps from semi-supervised data, while the self-training loss is used to balance the generator and discriminator to improve the model effectiveness.
- **SemiCDNet** [20]: SemiCDNet is a semi-supervised CD method using generative adversarial networks. The semi-supervised CD accuracy is improved by using two discriminators to enhance the consistency of the feature distribution of the segmentation and entropy graphs between labeled and unlabeled data thus improving the generalization ability of the generator by using a large amount of unlabeled data.

For the sake of fairness, the supervised models in the comparison models are trained using only labeled data.

### 4.4.1. Prediction on LEVIR2000 Dataset

The quantitative results of the proposed method and baseline on the LEVIR2000 dataset are shown in Table 2. Based on different sampling rate settings of labeled data, we calculated and summarized three evaluation metrics, F1-score, Kappa, and IoU.

It can be seen from Table 2 that our semi-supervised model on the LEVIR2000 dataset is significantly better than the fully supervised method and other semi-supervised methods. Among them, s4GAN and semiCDNet, which are also semi-supervised methods, do not have much improvement in model performance in this experiment compared to fully supervised methods when the sampling rate of labeled data is 2.5%. There are two possible reasons for this. On the one hand, our sampling rate is lower than the sampling rate in the original paper [20], which makes it difficult to fully learn the distribution of the dataset. On the other hand, the original paper [20] only uses remote sensing images containing changed pixels. As we know, GAN aims to train the generator to treat the discriminator. In the semi-supervised change detection method based on GAN, the discriminator should determine which are labels and which are predictions output by the model. The use of unchanged images means we used the all-black image as a label. This may lead to the result that the generator treats the discriminator by outputting a black image. This can be a limitation of semi-supervised change detection based on GAN. In our experiment, we

still use the unchanged images because we think unchanged is also a kind of label. This causes the performance of GAN to deteriorate in the later stages of training. However our methods can overcome the influence of using unchanged images. Additionally, we believe it is more convenient for practical applications to not distinguish between changed and unchanged images when selecting training data. It can be seen in Table 2 that our method significantly outperforms the comparison methods in every index. We will discuss the effectiveness and limitations of our method in detail in Section 4.5.1.

**Table 2.** The experimental results on the LEVIR2000 dataset.

| Method | Labeled Ratio | | | | | | | | | | | | | | | |
| --- | --- | --- | --- | --- | --- | --- | --- | --- | --- | --- | --- | --- | --- | --- | --- | --- |
| | 2.5% | | | | 5% | | | | 10% | | | | 20% | | | |
| | F1 | OA | Kappa | IoU | F1 | OA | Kappa | IoU | F1 | OA | Kappa | IoU | F1 | OA | Kappa | IoU |
| Fc-Diff | 0.2801 | 0.9416 | 0.2497 | 0.1629 | 0.2846 | 0.9570 | 0.2660 | 0.1659 | 0.3553 | 0.9536 | 0.3319 | 0.2160 | 0.6261 | 0.9728 | 0.6123 | 0.4557 |
| Fc-Conc | 0.4702 | 0.9592 | 0.4491 | 0.3074 | 0.5092 | 0.9632 | 0.4903 | 0.3415 | 0.6437 | 0.9725 | 0.6295 | 0.4746 | 0.7472 | 0.9807 | 0.7373 | 0.5964 |
| SNUNet-ECAM | 0.5398 | 0.9679 | 0.5238 | 0.3696 | 0.5787 | 0.9694 | 0.5632 | 0.4072 | 0.7410 | 0.9725 | 0.7306 | 0.5886 | 0.8010 | 0.9843 | 0.7929 | 0.6681 |
| s4GAN | 0.4430 | 0.9657 | 0.4277 | 0.2845 | 0.5654 | 0.9716 | 0.5520 | 0.3941 | 0.7728 | 0.9823 | 0.7784 | 0.6490 | 0.8076 | 0.9846 | 0.7995 | 0.6772 |
| semiCDNet | 0.5409 | 0.9657 | 0.5234 | 0.3707 | 0.6451 | 0.9736 | 0.6317 | 0.4762 | 0.7947 | 0.9840 | 0.7864 | 0.6593 | 0.8181 | 0.9852 | 0.8104 | 0.6922 |
| **semiSANet** | **0.7875** | **0.9835** | **0.7790** | **0.6494** | **0.8095** | **0.9852** | **0.8019** | **0.6800** | **0.8532** | **0.9881** | **0.8470** | **0.7440** | **0.8699** | **0.9895** | **0.8644** | **0.7698** |

To show this comparison more intuitively, we present the change detection maps of each model at a sampling rate of 10% in Figure 6.

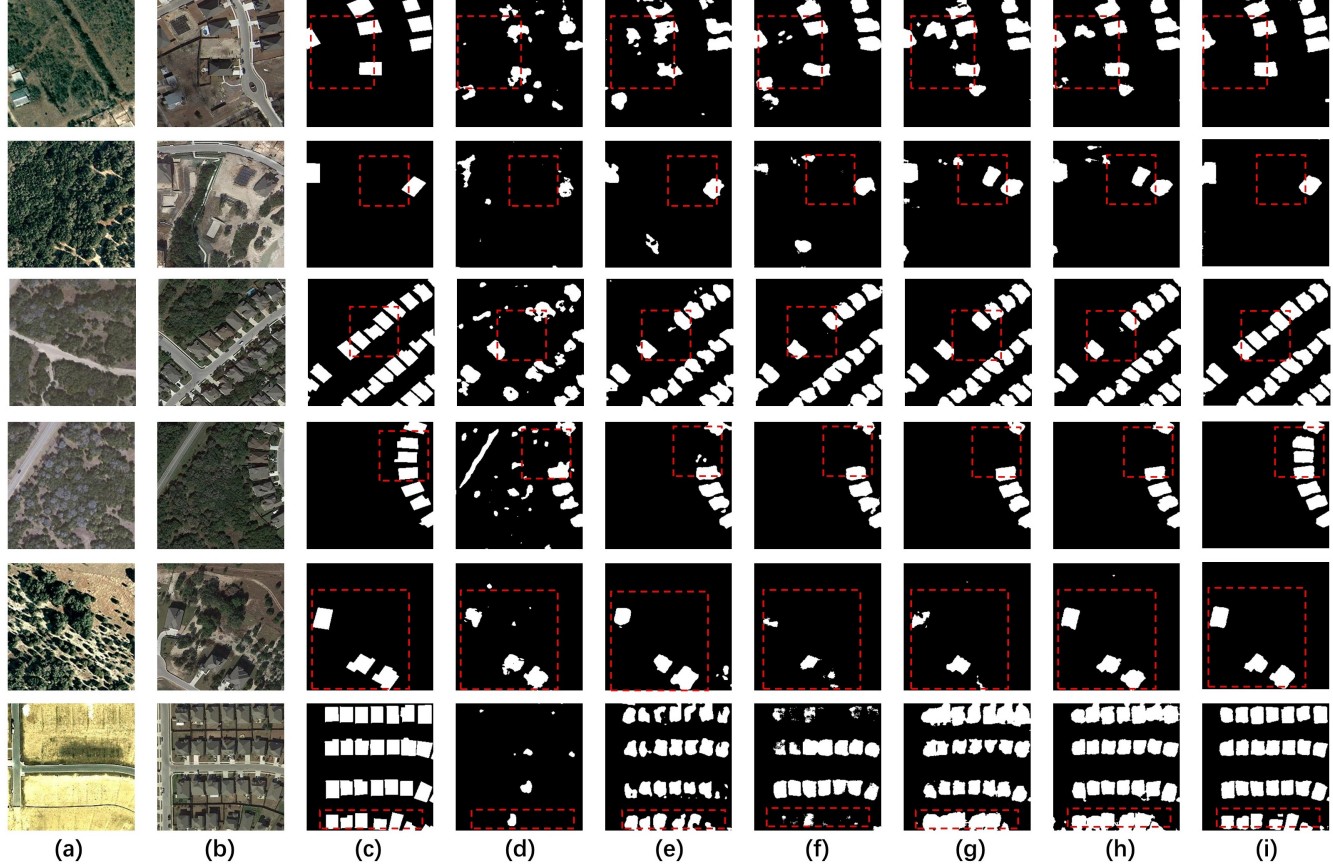

(a)     (b)     (c)     (d)     (e)     (f)     (g)     (h)     (i)

**Figure 6.** Visual comparisons of CD maps by different approaches on LEVIR2000 dataset using 10% labeled images for training. (**a**) Image $T_1$. (**b**) Image $T_2$. (**c**) Ground truth. (**d**) Fc-Diff. (**e**) Fc-Conc. (**f**) SNUNet-ECAM. (**g**) s4GAN. (**h**) semiCDNet. (**i**) semiSANet.

The first two rows of Figure 6 show the effective solution of our semi-supervised method for the false detection error. The comparison method judges the foundations of the buildings in the image as completed buildings, and our semiSANet identifies this kind of pseudo-change. The third and fourth rows of Figure 6 show that semiSANet is able to overcome the missed detection error. When the color of the changing building is similar to the color of the surrounding features, other methods judge this as background, i.e., no change has occurred. Additionally, our semiSANet successfully detects these changing objects. The reduction of false detection and missed detection errors is due to the strong augmentation method we use to improve the robustness of the model. The fifth and sixth rows of Figure 6 show that semiSANet is able to better restore the contours of pairs of changing objects and does not produce voids and blurs.

### 4.4.2. Prediction on WHU Dataset

The quantitative results of the proposed method and the baseline in the WHU dataset are shown in Table 3. Based on different sampling rate settings of labeled data, we calculated and summarized three evaluation metrics, F1-score, OA, Kappa and IoU.

As can be seen from Table 3, when the sampling rate is less than or equal to 20%, the results obtained by several fully supervised methods do not show an upward trend as the amount of labeled data increases. Additionally, the F1-score is always below 59%, Kappa is below 54%, and IoU is below 42%. This is due to the fact that the scene distribution of the WHU dataset is more complex than the LEVIR dataset and there are many pseudo-changes due to color changes. The model does not learn the overall distribution of the dataset when there is insufficient labeled data. Therefore, the model does not work well and is jittery when the sampling rate of labeled data is between 2.5% and 20%.

**Table 3.** The experimental results on the WHU dataset.

| Method | Labeled Ratio | | | | | | | | | | | | | | | |
| | 2.5% | | | | 5% | | | | 10% | | | | 20% | | | |
| | F1 | OA | Kappa | IoU | F1 | OA | Kappa | IoU | F1 | OA | Kappa | IoU | F1 | OA | Kappa | IoU |
|---|---|---|---|---|---|---|---|---|---|---|---|---|---|---|---|---|
| Fc-Diff | 0.4667 | 0.9177 | 0.4250 | 0.3044 | 0.4117 | 0.9122 | 0.3684 | 0.2592 | 0.4462 | 0.8250 | 0.3609 | 0.2872 | 0.4654 | 0.8322 | 0.3833 | 0.3032 |
| Fc-Conc | 0.3713 | 0.8089 | 0.2764 | 0.2280 | 0.4932 | 0.8678 | 0.4233 | 0.3273 | 0.3702 | 0.7248 | 0.2594 | 0.2272 | 0.3969 | 0.7324 | 0.2902 | 0.2476 |
| SNUNet-ECAM | 0.5407 | 0.8809 | 0.4776 | 0.3705 | 0.5175 | 0.8476 | 0.4432 | 0.3491 | 0.5830 | 0.9118 | 0.5342 | 0.4115 | 0.4467 | 0.7960 | 0.3552 | 0.2875 |
| s4GAN | 0.4624 | 0.8172 | 0.3766 | 0.3007 | 0.4484 | 0.8022 | 0.3583 | 0.2890 | 0.4405 | 0.7842 | 0.3463 | 0.2824 | 0.4251 | 0.7894 | 0.3303 | 0.2699 |
| semiCDNet | 0.5401 | 0.8837 | 0.4780 | 0.3699 | 0.5864 | 0.8934 | 0.5279 | 0.4130 | 0.6247 | 0.9252 | 0.5833 | 0.4543 | 0.6025 | 0.9266 | 0.5622 | 0.4312 |
| **semiSANet** | **0.7808** | **0.9579** | **0.7576** | **0.6405** | **0.7944** | **0.9611** | **0.7729** | **0.6589** | **0.8353** | **0.9693** | **0.8184** | **0.7172** | **0.8786** | **0.9778** | **0.8664** | **0.7834** |

The semi-supervised methods s4GAN, semiCDNet, and semiSANet also perform very differently on the WHU dataset. s4GAN performs even worse than fully supervised methods on the WHU dataset. semiCDNet has a performance-enhancing effect on the model, and this effect increases with the growth of the amount of labeled data. When the sampling rate of the labeled data is 20%, i.e., using 400 pairs of labeled images, semiCDNet results in an improvement of F1-score, Kappa, and IoU by about 15% over supervised methods. However, it is still not very satisfactory, with an F1-score of 60.25%, Kappa of 56.22%, and IoU of 43.12%. There are two main reasons for the poor realizations of these two semi-supervised methods. The first is the WHU dataset is challenging, with complex scenes, diverse change objects, and many pseudo-changes. The second is that we not only use the unchanged remote sensing image pairs as a kind of labeled data, but also use a lower sampling rate than the original paper [20]. As we have analyzed in Section 4.4.1, this reason can lead to the deterioration of GAN.

Our semiSANet works best on the WHU dataset, with a significant improvement in model performance. It shows the same trend as semiCDNet, i.e., the performance of the

model is enhanced when the number of labeled images increases, rather than jittering as in the fully supervised method. semiSANet works well at all sampling rates, with an F1-score more than 78%, Kappa more than 75%, and IoU more than 64%. When the sampling rate is 20%, semiSANet can achieve excellent performance with an F1-score of 87.86%, Kappa of 86.64%, and IoU of 78.34%.

To see more intuitively the improvement of model performance by our semi-supervised approach, we present the change detection results of each model at a sampling rate of 10% in Figure 7.

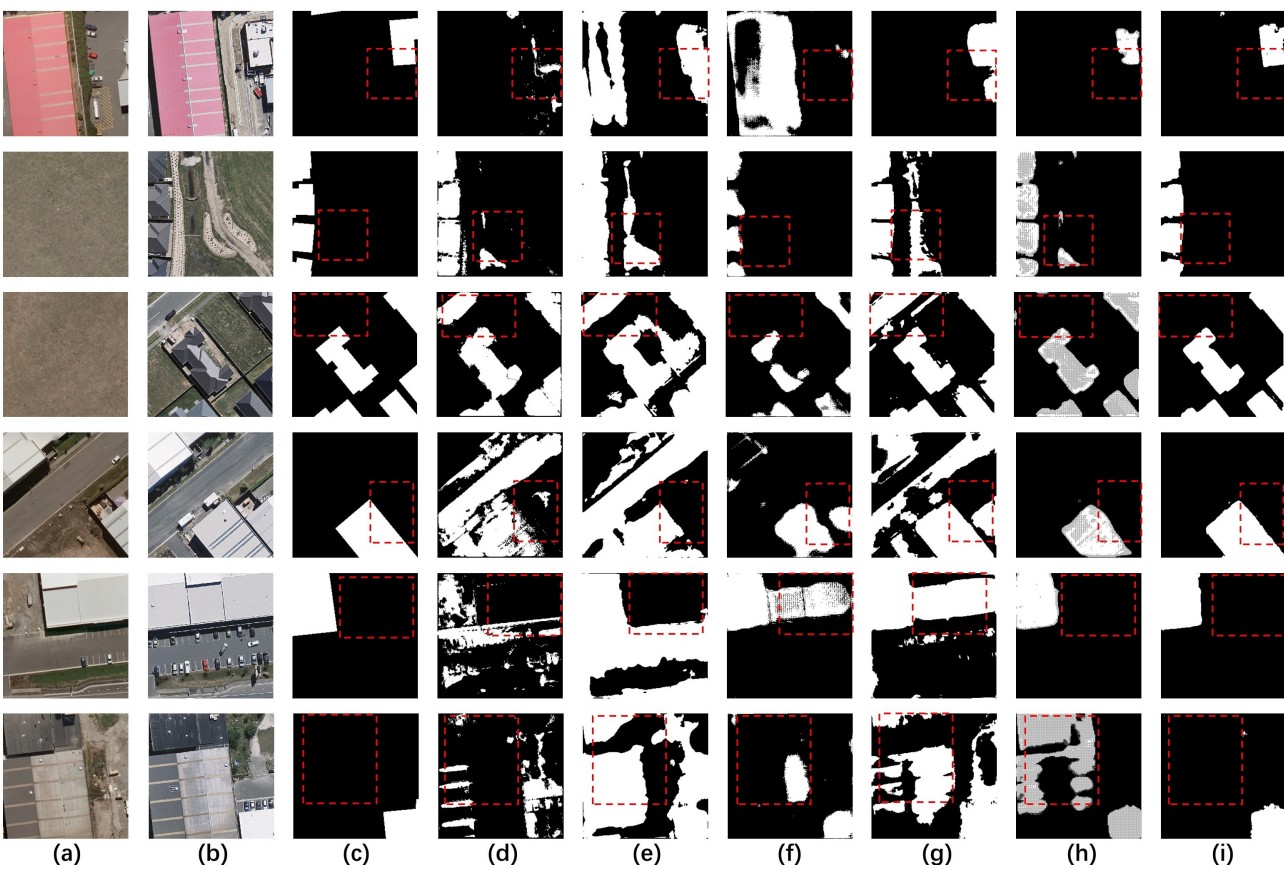

**Figure 7.** Visual comparisons of CD maps by different approaches on WHU dataset using 10% labeled images for training. (**a**) Image $T_1$. (**b**) Image $T_2$. (**c**) Ground truth. (**d**) Fc-Diff. (**e**) Fc-Conc. (**f**) SNUNet-ECAM. (**g**) s4GAN. (**h**) semiCDNet. (**i**) semiSANet.

From Figure 7, we can see that there are many false detections in the comparison methods, and these false detections mainly originate from the pseudo-changes in the remote sensing image pairs. Compared with the LEVIR2000 dataset, the change objects in the WHU dataset have more diverse shapes, sizes, and colors. Many of the changed houses do not have a clear sense of three-dimensionality on the map, just like a piece of land or the foundation of a building. These are the reasons for the confusion about the model causing poor change detection. However, our semi-supervised method is able to obtain good results even with only a small number of labels, because the robustness of the model is strengthened by the use of strong augmentation and consistency regularization, which makes the model less disturbed by these colors and shapes.

### 4.5. Ablation Studies

To better determine the effects of each component of the model on the effectiveness of the CD task, we designed ablation experiments to explore the effects of the semi-supervised method, semi-supervised loss weights $\lambda$, confidence threshold $\tau$, the amount of unlabeled data, and the graph attention module on the model.

### 4.5.1. Effect of the Semi-Supervised Method

To demonstrate the effectiveness of the semi-supervised algorithm, we conducted experiments using the fully supervised algorithm SANet and using the semi-supervised algorithm semiSANet at each labeled data sampling ratio.

The experimental results on the LEVIR2000 are shown in Table 4.

**Table 4.** The experimental results on the LEVIR2000 dataset.

| Method | Labeled Ratio | | | | | | | | | | | | | | | |
|--------|--------|--------|--------|--------|--------|--------|--------|--------|--------|--------|--------|--------|--------|--------|--------|--------|
| | 2.5% | | | | 5% | | | | 10% | | | | 20% | | | |
| | F1 | OA | Kappa | IoU | F1 | OA | Kappa | IoU | F1 | OA | Kappa | IoU | F1 | OA | Kappa | IoU |
| SANet | 0.5399 | 0.9682 | 0.5243 | 0.3698 | 0.6548 | 0.9745 | 0.6419 | 0.4868 | 0.7872 | 0.9831 | 0.7784 | 0.6490 | 0.8139 | 0.9848 | 0.8060 | 0.6862 |
| semi-SANet | 0.7875 | 0.9835 | 0.7790 | 0.6494 | 0.8095 | 0.9852 | 0.8019 | 0.6800 | 0.8532 | 0.9881 | 0.8470 | 0.7440 | 0.8699 | 0.9895 | 0.8644 | 0.7698 |

When the sampling rate is 2.5%, i.e., using only 50 labeled images, our model improves the F1-score from 53.99% to 78.75%, the Kappa from 52.38% to 77.90%, and the IoU from 36.96% to 64.94%, all by more than 24%. When the sampling rate is 5%, i.e., using 100 labeled images, our model improves the F1-score from 65.48% to 80.95%, improves Kappa from 64.19% to 80.19%, and improves IoU from 48.68% to 68.00%, all having more than 15% improvement. When the sampling rate is 10%, i.e., when 200 labeled images are used, the semi-supervised method improves the F1-score by 6.6%, the Kappa by 6.86%, and the IoU by 9.5%. When the sampling rate is 20%, i.e., the number of labeled images is 400, the semi-supervised method can improve the F1-score, Kappa, and IoU by 5.6%, 5.84%, and 8.36%, respectively.

The experimental results demonstrate the effectiveness of our semi-supervised method, whose improvement in model performance on the LEVIR2000 dataset is significant compared to using only labeled data. Additionally, the improvement is more obvious when the amount of labeled data is smaller. This is due to the fact that the robustness and generalization of the model are greatly improved by using strong augmentation to enhance the consistency of the model for unsupervised data change detection. Even with little labeled data, the model can improve robustness and generalization by using unlabeled data to enhance resistance to interference, including color change interference and shape change interference, thus greatly improving the change detection performance with little labeled data.

To see this improvement more visually, we show the fully supervised and semi-supervised results at a sampling rate of 2.5% and 5% in Figure 8. From Figure 8, we can see that when only 50 labeled images are used, the fully supervised results are very poor and almost impossible to detect the changed objects. The semi-supervised method adds another 1950 unlabeled images to it, and the detection effect is significantly improved, and all the changed objects are successfully detected. When only 100 labeled images are used, the fully supervised SANet detects some of the changed objects, but the detection results are very rough. Using the semi-supervised method with another 1900 unlabeled images, not only all objects are detected, but also the detection results are more accurate and have clear contours.

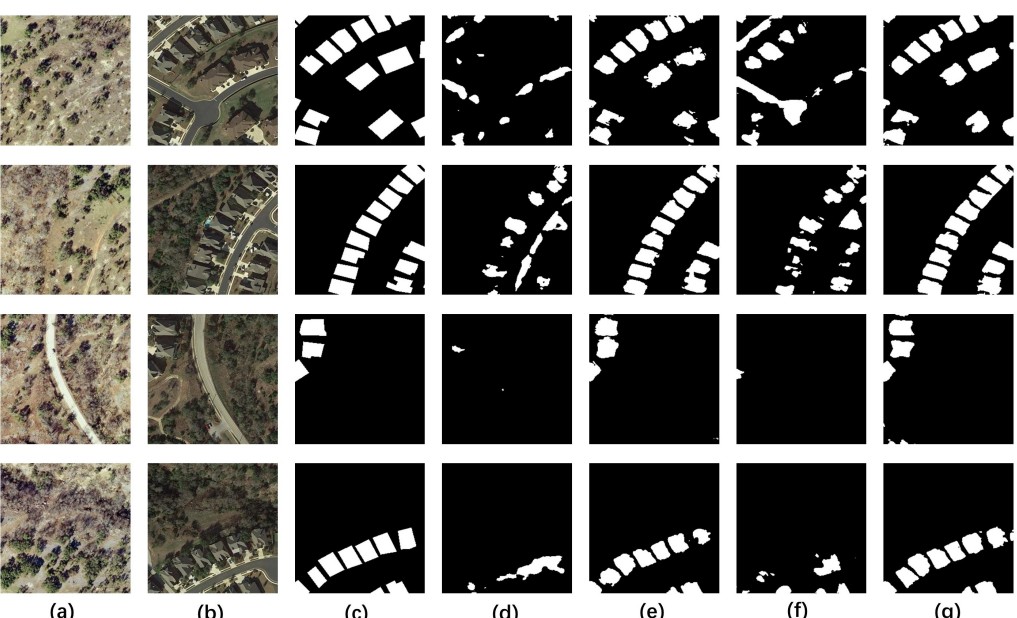

**Figure 8.** Visual comparison of SANet and semiSANet for CD maps on the LEVIR2000 dataset. (**a**) Image $T_1$. (**b**) Image $T_2$. (**c**) Ground Truth. (**d**) SANet with 50 labeled images. (**e**) semiSANet with 50 labeled images. (**f**) SANet with 100 labeled images. (**g**) semiSANet with 100 labeled images.

We conducted a comparison experiment between SANet and semiSANet on the WHU dataset and the results are shown in Table 5. It can be seen that using the semi-supervised algorithm improves the model performance significantly at each sampling rate. The improvement in F1-score is over 20%, and the improvement in Kappa and IoU are both over 25%. When the sampling rate is 20%, the results of semiSANet are excellent. The improvement of F1-score, Kappa, and IoU is all over 36%.

**Table 5.** The experimental results for the WHU dataset.

| Method | Labeled Ratio | | | | | | | | | | | | | | | |
|---|---|---|---|---|---|---|---|---|---|---|---|---|---|---|---|---|
| | 2.5% | | | | 5% | | | | 10% | | | | 20% | | | |
| | F1 | OA | Kappa | IoU | F1 | OA | Kappa | IoU | F1 | OA | Kappa | IoU | F1 | OA | Kappa | IoU |
| SANet | 0.5651 | 0.8846 | 0.5044 | 0.3938 | 0.4644 | 0.8313 | 0.3820 | 0.3024 | 0.5693 | 0.8798 | 0.5073 | 0.3979 | 0.5151 | 0.8682 | 0.4464 | 0.3469 |
| semi-SANet | 0.7808 | 0.9579 | 0.7576 | 0.6405 | 0.7944 | 0.9611 | 0.7729 | 0.6589 | 0.8353 | 0.9693 | 0.8184 | 0.7172 | 0.8786 | 0.9778 | 0.8664 | 0.7834 |

To see this improvement more intuitively, we show the fully supervised and semi-supervised results in Figure 9 for sampling rates of 2.5% and 5%. From Figure 9 it can be seen that there are many false detections in the results when only the fully supervised algorithm is used. This is because, in the WHU dataset, there are many geographical objects around the building that are similar in color to the building, such as roads. The semi-supervised algorithm can eliminate these false detections very well. This is because when we use strong augmentation on the image, the color gap between different geographic objects is sometimes magnified and sometimes minimized. The consistency regularization makes the model produce the same change detection results for both cases where the color of the objects are similar and are very different. Eventually, the model will focus more on the semantic gaps of the objects rather than the color gaps.

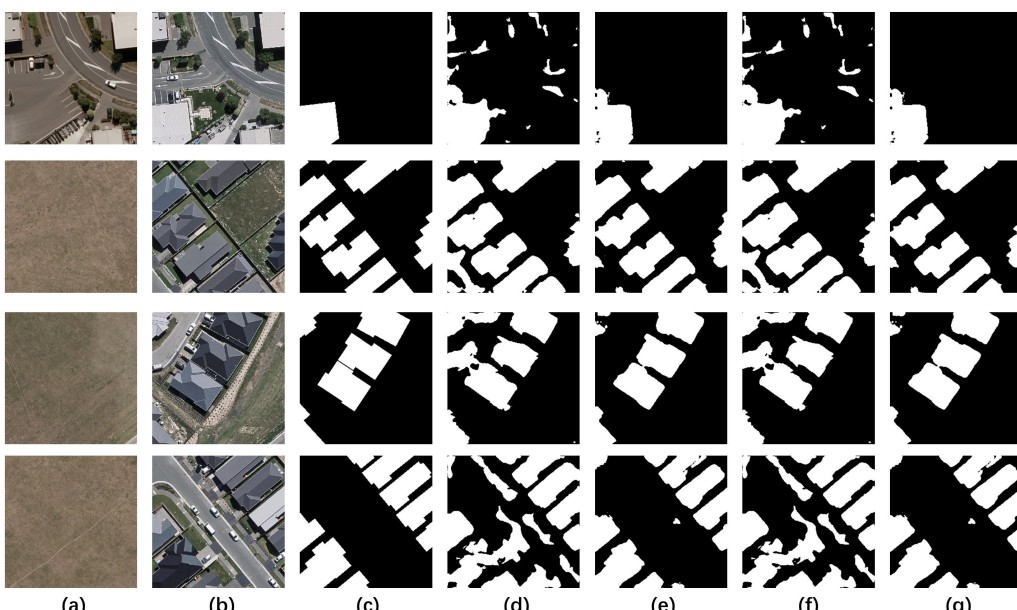

**Figure 9.** Visual comparison of SANet and semiSANet methods for CD maps for the WHU dataset. (**a**) Image $T_1$. (**b**) Image $T_2$. (**c**) Ground truth. (**d**) SANet with 50 labeled images. (**e**) semiSANet with 50 labeled images. (**f**) SANet with 100 labeled images. (**g**) semiSANet with 100 labeled images.

4.5.2. Effect of Unsupervised Loss Weights $\lambda$

The size of $\lambda$ determines how much the unlabeled image affects the model effect. To find the optimal range of values for $\lambda$ on semiSANet, we set the confidence threshold $\tau = 0.95$ and $\lambda = \{0, 0.05, 0.5, 1, 2, 4, 8, 12, 16, 32, 64\}$ for experiments on LEVIR2000 dataset and WHU dataset. The experimental results are shown in Figure 10. From Figure 10a, we can see that the effect of change detection rises rapidly and then decreases slowly when $\lambda$ increases on the LEVIR2000 dataset. In our experiments, $\lambda \in [0, 2]$ is the fast rising stage, increasing the value of $\lambda$ at this stage can significantly improve the model effect. The model effect is satisfactory when $\lambda \in [4, 16]$, and the F1-score is stable at about 85%, and reaches the highest value of 85.32% when $\lambda = 8$. When $\lambda > 16$, the model effect slowly decreases, but it is still much better than the fully supervised effect. From Figure 10b, the same trend is seen on the WHU data set. The $\lambda \in [0, 4]$ is the fast-rising phase. After that, the F1-score increases slowly and gets the maximum value of 84.19% at $\lambda = 16$. Then the F1-score decreases slowly with the increase of $\lambda$. This experiment illustrates the large range of effective values for the unsupervised loss weights $\lambda$, probably because the unsupervised loss $L_u$ is much smaller than the supervised loss $L_s$, so a large value of $\lambda$ is not enough to make $L_u$ destroy the guiding effect of the supervised loss $L_s$ on the model. However, if the value of $\lambda$ is taken too small, the contribution to the model effect is weak. In addition, taking values within the interval $[4, 20]$ may give the best results.

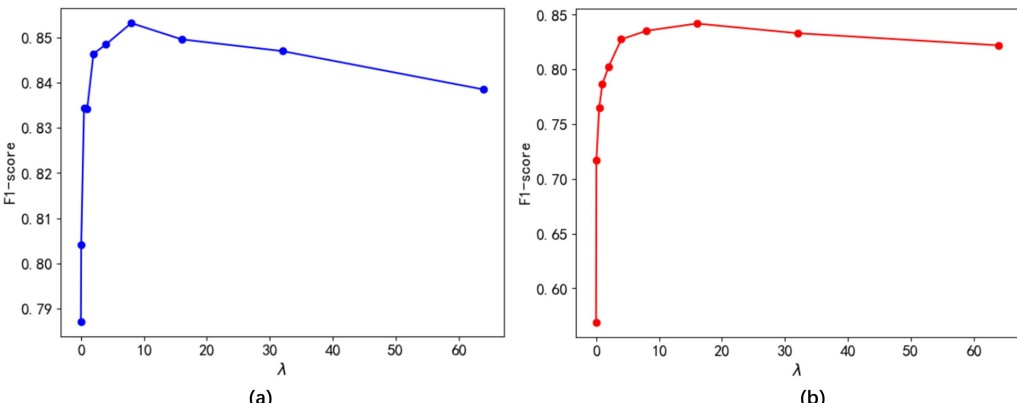

**Figure 10.** Effects of semi-supervised loss weights $\lambda$. (**a**) LEVIR2000 dataset. (**b**) WHU dataset.

### 4.5.3. Effect of Confidence Threshold $\tau$

The value of $\tau$ determines how many pixels in the pseudo-label are utilized to compute the loss, which has an important impact on the model effect. In order to find the best value range of $\tau$ on semiSANet, we set $\lambda = 8$ and $\tau = \{0.5, 0.6, 0.7, 0.8, 0.9, 0.95, 0.99\}$ and conduct experiments on LEVIR2000 dataset and WHU dataset. The experimental results are shown in Figure 11. From Figure 11a, the model performance does not change much on the LEVIR2000 dataset. The F1-score jittered between 84.94% and 85.39% when $\tau \in [0.5, 0.95]$, and only when $\tau = 0.99$, did the F1-score drop to 84.39%. This may be due to the fact that the model was pre-trained well on the LEVIR2000 dataset as shown in Table 4, so it can still get the pseudo-label well even when the confidence threshold $\tau$ takes a small value. However, when $\tau$ is too large, pseudo-labels that are useful for model training are filtered out instead, leading to a drop in model performance.

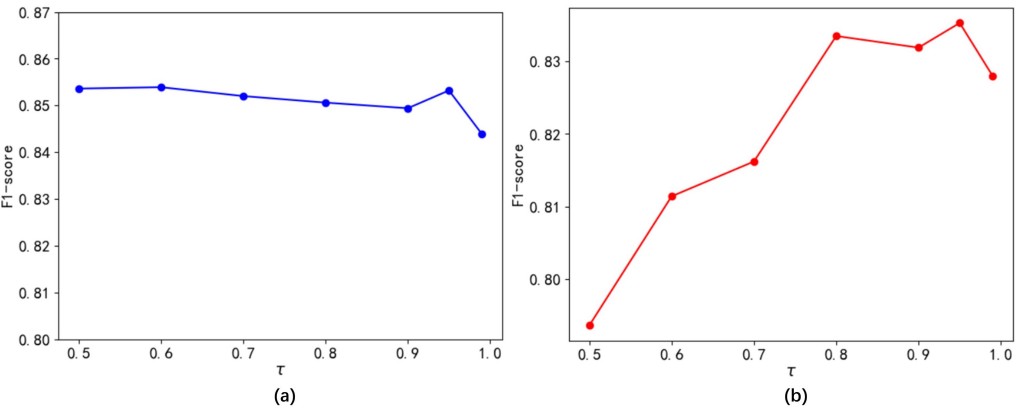

**Figure 11.** Effects of confidence threshold $\tau$. (**a**) LEVIR2000 dataset. (**b**) WHU dataset.

From Figure 11b, it can be seen that the F1-score shows a trend of first increasing and then decreasing with the growth of $\tau$ on the WHU dataset. This is due to the poor effectiveness of the pre-trained model on the WHU dataset leading to the presence of many false labels in the pseudo-labels which must be filtered out with the confidence threshold $\tau$. It can be seen that when $\tau \in [0.5, 0.8]$, the F1-score gradually increases from 79.36% to 83.35% as $\tau$ grows and the wrong pseudo-label is gradually filtered out. When $\tau \in [0.8, 0.95]$, the model is in a more stable stage, and the F1-score is jittering between 83.19% and 83.53%. However, when $\tau = 0.99$, the F1-score drops to 82.8%. This experiment illustrates that the confidence threshold $\tau$ should be taken with reference to the goodness of the pre-training effect. In general the model achieves better results when $\tau \in [0.8, 0.95]$.

### 4.5.4. Effect of Unlabeled Data Volume

In order to test whether the increase in the amount of unlabeled data has a contributing effect on the improvement of semi-supervised performance, we conducted experiments on the LEVIR dataset and the WHU dataset. Previously, we used only 2000 pairs of images from the LEVIR dataset as the training set, but in this experiment we used up to 4000 images from the LEVIR dataset as the training set in order to test the effect of a larger amount of unsupervised data. We set the number of training sets for the LEVIR dataset to {200, 400, 600, 1000, 2000, 3000, 4000}, where the amount of labeled data is 200. We set the number of training sets for the WHU dataset to {200, 400, 600, 1000, 1500, 2000}, where the amount of labeled data is also 200.

The experimental results are shown in Figure 12. Overall, the change detection effectiveness of the model tends to increase as the amount of unsupervised data increases. From the results of LEVIR in Figure 12a, there is also a jitter in this rising process, and not every increase in the amount of unsupervised data has improved the model performance. Additionally, when the number of training sets is only 400, the semi-supervised method already has a large improvement for the model performance. This may be because 200 pairs of images as labeled data can already represent the overall distribution of the LEVIR dataset well, and only 200 more pairs of unlabeled images need to be added to improve the robustness and generalization of the model to obtain a higher change detection accuracy. The results for the WHU dataset in Figure 12b show the same trend. The model performance has a large improvement when the number of training sets is 400, after which the F1-score slowly improves, but due to the limitation of data volume, we cannot observe the situation of the WHU dataset after the data volume exceeds 2000.

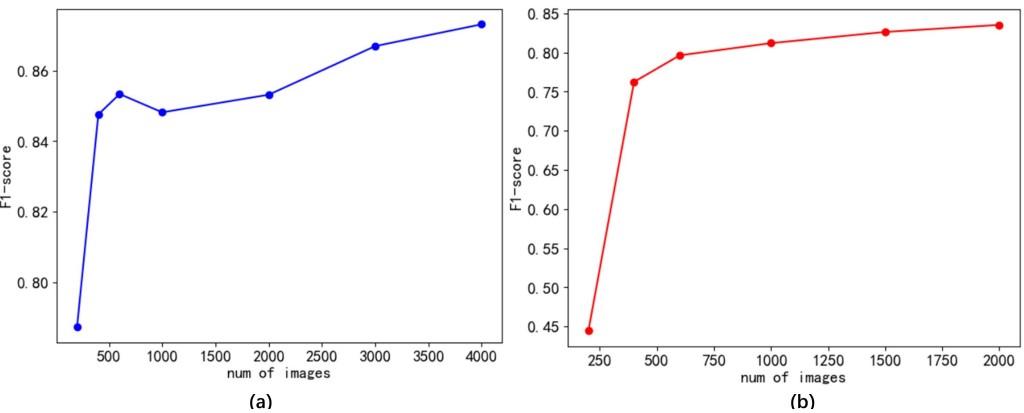

**Figure 12.** Effects of the data volume. (**a**) LEVIR dataset. (**b**) WHU dataset.

### 4.5.5. Effect of the GAT Module

To make the model have the ability to compute feature correlation and fuse long-distance information, we added the GAT module at the end of the encoder in SANet. To verify the usefulness of the GAT module, we did ablation experiments on the LEVIR2000 dataset and the WHU dataset with a 10% sampling rate of labeled images and the Google dataset, and the results are shown in Figure 13. The results demonstrate that the GAT module can indeed improve the performance of the model. Adding GAT to the LEVIR2000 dataset with a sampling rate of 10% resulted in a 0.88%, 0.9% and 1.33% improvement in F1-score, Kappa, and IoU, respectively. Adding the GAT on the WHU dataset with a 10% sampling rate resulted in 0.96%, 1.13%, and 1.4% improvements in F1-score, Kappa, and IoU, respectively.

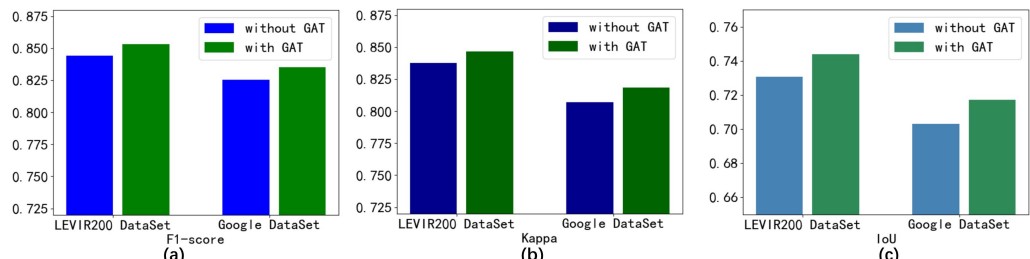

**Figure 13.** Effects of the GAT module. (**a**) F1-score. (**b**) Kappa. (**c**) IoU.

## 5. Discussion

The proposed semi-supervised change detection method can achieve effective change detection results in case of insufficient labeled images. Firstly, the proposed method is simple. Inspired by FixMatch [47], this work achieves semi-supervised change detection by only using strong augmentation and confidence threshold filter. Compared with semi-supervised remote sensing image processing methods using mean-teacher [19] or GAN [20,49], this work is easier to be implemented. Moreover, extensive experiments prove that this work is more effective than previous work. With small amount of labeled data, this work achieves better change detection results than the state-of-the-art supervised methods and semi-supervised methods.

However, there are two main limitations in this work. First, the proportion of unlabeled data is limited by GPU memory. As mentioned in Section 4.3, the batch size of the labeled data should be 1 at least, and the batch size of unlabeled data will be $\frac{1}{k} - 1$, where $k$ means the proportion of labeled data. With the proportion of unlabeled data increasing, the batch size of input data will increase and require more GPU memory. Second, strong augmentation may destroy the useful information of low and middle-resolution remote sensing images. This work aims at high resolution remote sensing images change detection, and strong augmentation is the most important technology in this work. However, when it comes to low and middle-resolution images which are coarser and easy to be disturbed, strong augmentation may destroy their semantic information.

## 6. Conclusions

In this paper, we propose a semi-supervised remote sensing images change detection method based on Siamese UNet and consistency regularization, called semiSANet. In semiSANet, we use a graph attention mechanism to fuse the information between extracted deep semantic features, which enables the model to fuse long-range feature information. To exploit the unlabeled data, we use the consistency regularization principle, i.e., to make the model produce the same change detection results for images before and after distortion. Specifically, the unlabeled data are first fed into the pre-trained model and confidence threshold filter to obtain pseudo-labels with high confidence. Meanwhile, the distorted images are obtained using strong augmentation, and the CD results of the distorted images are obtained by the model. By reducing the difference between CD results and pseudo-label, the interference caused by strong augmentation is mitigated and the robustness of the model is improved. The experimental results on two high-resolution remote sensing image datasets demonstrate that our method can significantly improve the performance with only a small amount of labeled data and outperforms the state-of-the-art methods. When there are only 2.5% labeled images, our methods can increase the IoU by more than 25% compared to the state-of-the-art methods.

In the future, we will focus on solving the limitations of this work. Specifically, the first one is to control the requirement of GPU memory, the batch size of labeled and unlabeled data should be fixed rather than change with the proportion of labeled data. Second, we will study how the strong augmentation range affects different resolution remote sensing image change detection.

**Author Contributions:** Conceptualization, C.S. and H.C.; methodology, C.S. and H.C.; software, C.S.; validation, C.S., C.D. and H.C.; data curation, H.C.; writing—original draft preparation, C.S.; writing—review and editing, C.D. and J.W.; supervision, H.C.; project administration, H.C. All authors have read and agreed to the published version of the manuscript.

**Funding:** The work in this paper is supported by the National Natural Science Foundation of China (41871248, 41971362, U19A2058).

**Data Availability Statement:** Not applicable.

**Conflicts of Interest:** The authors declare no conflict of interest.

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
