# Peer review of "SemiSANet: A Semi-Supervised High-Resolution Remote Sensing Image Change Detection Model Using Siamese Networks with Graph Attention"

_remotesensing, doi:10.3390/rs14122801_

Round 1

Reviewer 1 Report

This is potentially a useful work but I had difficulty to understand so many elements of the work given that computing people wrote it. Hence, remote sensing part is not well organised nor explained clearly. Further, clarification is albeit needed to present remote sensing part and lay down how this study contributes to knowledge relative to what is already known. For example see this work https://doi.org/10.1016/j.array.2021.100102. International significance of the study is also lacking as it currently stands. Some comments are appended blow.

[1] When you convey results of your work in the abstract, use paste tense. There are many things in your abstract which are obscure and not providing clear info to readers. For example, what do you mean by ‘labelled dual-temporal remote sensing’? This type of expressions are widespread in the work, making it hard to understand.

[2] Introduction section is poorly handled motivation and novelty of this work. Change detection can be done from tabula data of two times or land use maps of two periods. But your first line indicates without remote sensing CD is not possible. You used many acronyms here but they have not been elaborated or utilise properly. Intro section needs to show how this work could add value and knowledge to what is already known by the scientific community

[3] Methods section is largely dealt with computing science algorithm, little is done for remote sensing science. This needing reorganisation and adding texts that are easy to use by remote sensing people. Above work could be useful to reorganise this section. What is WHU dataset? How do I understand this?

[4] Discussion section: should be improved by using state-of-the art knowledge to revamp it for scientific community especially remote sensing audience.

[5] English needs significant improvement and correction

Reviewer 2 Report

Dear Authors,

Thank you for submitting your paper to Remote Sensing Journal. This works seems interesting but some modifications are required for publication.

  1. Abstract should included the numerical i.e. the percentage how much was improved with your method.
  2. A chapter on related work is interesting but introduction can be improved by adding the works on other semi-supervised methods. Some of the papers are:  https://doi.org/10.3390/rs14051264 , https://doi.org/10.3390/rs12091528 , https://doi.org/10.1016/j.neucom.2021.12.077, DOI: 10.1109/IGARSS47720.2021.9554652.
  3. Overall Accuracy was shown in equation but the values are not tabulated. Please add those values too for better evaluation.
  4. Please elaborate how the confidence threshold filter was decided.
  5. Please add the numericals in conclusion too.

Thank you.

Reviewer 3 Report

Manuscript " SemiSANet: A Semi-supervised High-Resolution Remote Sensing Image Change Detection Model using Siamese Networks with Graph Attention." The authors proposed a semi-supervised change detection method based on consistency regularization and strong augmentation. Firstly, I have a few doubts about the novelty of this method. Please check comment 1.

Secondly, the scientific language of this work is so poor, with so many flaws and vague statements. It is recommended to ask a native expert to polish the English for this work gently.

Comment 1.

Please clarify what this work/methodology is making different than

  1. Peng, D., et al., SemiCDNet: A semisupervised convolutional neural network for change detection in high-resolution remote-sensing images. IEEE Transactions on Geoscience and Remote Sensing, 2020. 59(7): p. 5891-5906. 10.1109/TGRS.2020.3011913

https://ieeexplore.ieee.org/document/9161009

  1. Jiang, X. and H. Tang. Dense high-resolution Siamese network for weakly-supervised change detection. In 2019 6th International Conference on Systems and Informatics (ICSAI). 2019. IEEE.

https://ieeexplore.ieee.org/document/9010267

  1. Wang, J., et al., Semi-supervised remote sensing image semantic segmentation via consistency regularization and average update of pseudo-label. Remote Sensing, 2020. 12(21): p. 3603.

Check their work carefully and clarify the novelty and what research gaps remained in their work, which the authors have addressed in this approach.

Comment 2.

It's strongly recommended to add line numbers in your manuscript. No one has spare time to count lines and paragraphs to write a review report.

Comment 3.

The introduction section of this work is written so poorly that long sentences and basic information without any reference make it duller (e.g.) 2nd paragraph. It is strongly recommended to rearrange the whole introduction section and maintain the sequence of the main idea.

Any reference to the first method presented there?

"There are only a few SSL-based CD models nowadays," e.g.??

"Sohn et al. [18] of Google Research"???

Comment 4.

I couldn't find sufficient information about the limitations of previous work that the authors considered in this work. Then, how about the rules and limitations of this method/work.

Comment 5.

This research, especially the results sections, seems weaker without discussing and validating their results with published literature. It's strongly recommended to add a discussion section, strengthen your results, and validate previously published work. Also, add a limitation section under discussion. And conclude your research accordingly.

Currently, this work seems confusing, and hard to make any decision. After a gentle modification and English revisions, this work can be considered to write a detailed review report.

Reviewer 4 Report

You have spelled some but not all technical abbreviations. It would be helpful to do all of them for consistency and help those who are not familiar with the methods, models or variables. they include and not limited to UDA, MLP, CM, FC-EF, etc.

Reviewer 5 Report

This research work presented semi-supervised learning - for change detection in remote sensing images. This research discusses consistency regularization and strong augmentation in semi-supervised learning. The proposed model is applied to appropriate industry applications to demonstrate its robustness.

The semi-supervised learning is a well-established research domain with a fair amount of literature including:

Van Engelen, J. E., & Hoos, H. H. (2020). A survey on semi-supervised learning. Machine Learning109(2), 373-440.

Diaz-Pinto, A., Colomer, A., Naranjo, V., Morales, S., Xu, Y., & Frangi, A. F. (2019). Retinal image synthesis and semi-supervised learning for glaucoma assessment. IEEE transactions on medical imaging38(9), 2211-2218.

The manuscript is well written with strong experimental analysis.

It would help if the manuscript may include:

  1. a mathematical description/discussion on how semisupervised learning is established.
  2. more description of how the learning is controlled within the unsupervised learning phase.
  3. more discussion on the proportion of supervised learning samples vs unsupervised learning samples - not an exhaustive theoretical discussion but some pointers for future discussion as this article is a more application-oriented one but needs to address all that matters to the audience a bit more (e.g., how did you come to decision on splitting the learning samples in half?)

This manuscript is well-written and I trust it presents an interesting work to the remote sensing community.

Round 2

Reviewer 1 Report

Although authors addressed 5 points of my previous comments, they did not address the following in their revision. Therefore, I like to see the following is addressed to enhance international significance and improved readership. 

This is potentially a useful work but I had difficulty to understand so many elements of the work given that computing people wrote it. Hence, remote sensing part is not well organised nor explained clearly. Further, clarification is albeit needed to present remote sensing part and lay down how this study contributes to knowledge relative to what is already known. For example see this work https://doi.org/10.1016/j.array.2021.100102. International significance of the study is also lacking as it currently stands. Some comments are appended blow.

Reviewer 3 Report

The manuscript is now significantly improved and acceptable for publication. But I am a bit concerned about discussing the results and validation of findings with previously done studies. The number of citations to support the "Experiment" section is too low.

In my previous suggestion, I recommended adding a new section, "5. Discussion," along with "5.1. Rules/Limitations of this work and future aspects", but I couldn't find that information in the revised version. It's better to discuss your main finding by comparing it with previous publication's results and ease the readers by reporting the limitations of this work. (Especially the limitations of your work)

Currently, the revised version contains 22% Plagiarism which should be minimized, With a dominant content  4% from a previous publication of MDPI and 

Daifeng Peng, Lorenzo Bruzzone, Yongjun Zhang, Haiyan Guan, Haiyong Ding, Xu Huang. "SemiCDNet: A Semisupervised Convolutional Neural Network for Change Detection in High Resolution Remote-Sensing Images", IEEE Transactions on Geoscience and Remote Sensing, 2021
